# The role of ions in new-particle formation in the CLOUD chamber

Robert Wagner[1], Chao Yan[1], Katrianne Lehtipalo[1,2], Jonathan Duplissy[3], Tuomo Nieminen[4], Juha Kangasluoma[1], Lauri R. Ahonen[1], Lubna Dada[1], Jenni Kontkanen[1,5], Hanna E. Manninen[1,6], Antonio Dias[6,7], Antonio Amorim[7,8], Paulus S. Bauer[9], Anton Bergen[10], Anne-Kathrin Bernhammer[11], Federico Bianchi[1], Sophia Brilke[9,10], Stephany Buenrostro Mazon[1], Xuemeng Chen[1], Danielle C. Draper[12], Lukas Fischer[11], Carla Frege[2], Claudia Fuchs[2], Olga Garmash[1], Hamish Gordon[6,13], Jani Hakala[1], Liine Heikkinen[1], Martin Heinritzi[10], Victoria Hofbauer[14], Christopher R. Hoyle[2], Jasper Kirkby[6,10], Andreas Kürten[10], Alexander N. Kvashnin[15], Tiia Laurila[1], Michael J. Lawler[12], Huajun Mai[16], Vladimir Makhmutov[15,17], Roy L. Mauldin III[1,18], Ugo Molteni[2], Leonid Nichman[19,20,21], Wei Nie[1,22], Andrea Ojdanic[9], Antti Onnela[6], Felix Piel[10,23], Lauriane L. J. Quéléver[1], Matti P. Rissanen[1], Nina Sarnela[1], Simon Schallhart[1], Kamalika Sengupta[13], Mario Simon[10], Dominik Stolzenburg[9], Yuri Stozhkov[15], Jasmin Tröstl[2], Yrjö Viisanen[24], Alexander L. Vogel[2,6], Andrea C. Wagner[10], Mao Xiao[2], Penglin Ye[14,20], Urs Baltensperger[2], Joachim Curtius[10], Neil M. Donahue[14], Richard C. Flagan[16], Martin Gallagher[19], Armin Hansel[11,23], James N. Smith[4,12], António Tomé[7], Paul M. Winkler[9], Douglas Worsnop[1,3,20,25], Mikael Ehn[1], Mikko Sipilä[1], Veli-Matti Kerminen[1], Tuukka Petäjä[1] and Markku Kulmala[1]

[1]Department of Physics, University of Helsinki, Helsinki, Finland
[2]Paul Scherrer Institute, Laboratory of Atmospheric Chemistry, Villigen, Switzerland
[3]Helsinki Institute of Physics, University of Helsinki, P.O. Box 64, Helsinki, Finland
[4]University of Eastern Finland, Department of Applied Physics, P.O. Box 1627, Kuopio, Finland
[5]Department of Environmental Science and Analytical Chemistry (ACES) & Bolin Centre for Climate Research, Stockholm University, Stockholm, Sweden
[6]CERN, Geneva, Switzerland
[7]CENTRA - SIM, University of Lisbon and University of Beira Interior, Lisbon, Portugal
[8]Faculty of Science and Technology, New University of Lisbon, Lisbon, Portugal
[9]University of Vienna, Faculty of Physics, Vienna, Austria
[10]Goethe University Frankfurt, Institute for Atmospheric and Environmental Sciences, Frankfurt am Main, Germany
[11]Institute for Ion and Applied Physics, University of Innsbruck, Innsbruck, Austria
[12]Department of Chemistry, University of California, Irvine, CA, USA
[13]University of Leeds, School of Earth and Environment, Leeds, United Kingdom
[14]Center for Atmospheric Particle Studies, Carnegie Mellon University, 5000 Forbes Ave, Pittsburgh, PA, USA
[15]Lebedev Physical Institute, Russian Academy of Sciences, Moscow, Russia
[16]California Institute of Technology, Pasadena, CA, USA
[17]Moscow Institute of Physics and Technology (State University), Moscow, Russia
[18]Department of Atmospheric and Oceanic Sciences, Boulder, Colorado
[19]School of Earth and Environmental Sciences, University of Manchester, Manchester, United Kingdom
[20]Aerodyne Research Inc., Billerica, MA, USA
[21]Department of Chemistry, Boston College, Chestnut Hill, MA, USA
[22]Joint International Research Laboratory of Atmospheric and Earth System Sciences, Nanjing University, Nanjing, China
[23]IONICON Analytik GmbH, Innsbruck, Austria
[24]Finnish Meteorological Institute (FMI), P.O. Box 503, Helsinki, Finland
[25]TOFWERK AG, Uttigenstrasse 22, Thun, Switzerland

*Correspondence to*: Markku Kulmala (markku.kulmala@helsinki.fi)

**Abstract.** The formation of secondary particles in the atmosphere accounts for more than half of global cloud condensation nuclei. Experiments at the CERN CLOUD (Cosmics Leaving OUtdoor Droplets) chamber have underlined the importance of ions for new particle formation, but quantifying their effect in the atmosphere remains challenging. By using a novel instrument setup consisting of two nano-particle counters, one of them equipped with an ion filter, we were able to further investigate the ion-related mechanisms of new particle formation. In autumn 2015, we carried out experiments at CLOUD on four systems of different chemical compositions involving monoterpenes, sulfuric acid, nitrogen oxides, and ammonia. We measured the influence of ions on the nucleation rates under precisely controlled and atmospherically relevant conditions. Our results indicate that ions enhance the nucleation process when the charge is necessary to stabilize newly formed clusters, i.e. in conditions where neutral clusters are unstable. For charged clusters that were formed by ion-induced nucleation, we were able to measure, for the first time, their progressive neutralization due to recombination with oppositely charged ions. A large fraction of the clusters carried a charge at 1.5 nm diameter. However, depending on particle growth rates and ion concentrations, charged clusters were largely neutralized by ion-ion recombination before they grew to 2.5 nm. At this size, more than 90% of particles were neutral. In other words, particles may originate from ion-induced nucleation, although they are neutral upon detection at diameters larger than 2.5 nm. Observations at Hyytiälä, Finland, showed lower ion concentrations and a lower contribution of ion-induced nucleation than measured at CLOUD under similar conditions. Although this can be partly explained by the observation that ion-induced fractions decrease towards lower ion concentrations, further investigations are needed to resolve the origin of the discrepancy.

## 1 Introduction

Aerosol particles influence our life in various ways by affecting our health, the water cycle and the global climate. The climate effect of aerosols is still poorly understood and contributes a large part of the uncertainty when estimating Earth's radiative forcing (IPCC, 2013). Aerosols can influence the radiative forcing directly by absorbing and scattering sunlight. Furthermore, when aerosol particles act as cloud condensation nuclei, they affect cloud brightness and lifetime (Albrecht, 1989). Besides direct emission from sources such as combustion processes, volcanoes or sea spray, aerosols are also produced in the atmosphere from condensable vapors via so-called new-particle formation (NPF; Kulmala et al. (2004)).

During the initial step of NPF, also known as particle nucleation, ions can play an important role by enhancing the stability of newly-formed molecular clusters (Yu and Turco, 2001) and reducing their evaporation rates. Key factors determining the influence of ions are the concentration of precursor vapors (Kulmala et al., 2014), the condensation sink of pre-existing particles (Kerminen et al., 2001; Kulmala et al., 2014), temperature (Kürten et al., 2016), and the ionization rate from galactic cosmic rays and terrestrial radioactivity such as radon (Zhang et al., 2011).

The term 'ion-induced nucleation' refers to the ion-assisted formation of thermodynamically stable particles, i.e. for which the growth rate exceeds the evaporation rate. Nucleation occurs at the critical size or, in the case of barrierless nucleation, upon dimer formation. The ion either directly stabilizes the molecular cluster or helps the embryonic charged cluster exceed the

stable size by recombination with an oppositely-charged cluster, which neutralizes the charge. To allow for the latter mechanism, Yu and Turco (2001), introduced the term 'ion-mediated nucleation'. Here we will refer to both processes collectively as ion-induced nucleation for consistency with earlier publications from the CLOUD project. Early laboratory measurements suggested that ion-induced nucleation of sulfuric acid particles would be important in the low temperatures of

the middle and upper troposphere, but not appreciable in the boundary layer (Lovejoy et al., 2004; Curtius et al., 2006). While some models predict a large contribution of ion-induced nucleation to aerosol particles in the global troposphere (Kazil et al., 2010; Yu et al., 2010), atmospheric observations in the boundary layer indicated only minor contributions from ion-induced nucleation (Gagne et al., 2008; Kontkanen et al., 2013; Kulmala et al., 2010; Kulmala et al., 2013; Manninen et al., 2010; Manninen et al., 2009). However, by using kinetic modelling and simplified analytical analysis of progressive

neutralization during particle growth, Yu and Turco (2011), provided a different interpretation of these atmospheric observations. They concluded that a major contribution of ion-induced nucleation cannot be ruled out, moreover, that the observations suggest that the ion-induced nucleation pathway may be dominant.

The signature of ion-induced nucleation in the atmosphere is the appearance and growth of charged molecular clusters just

above the size range of small ions. Here we will refer to particles measured above a certain detection threshold as particle *formation*, whereas we use particle *nucleation* to refer to the formation of thermodynamically stable particles above the critical size. Measurements in the boundary layer at the boreal forest site in Hyytiälä, Finland, suggested that ion-induced nucleation contributes around 10% to total new particle formation between 2–3 nm (Manninen et al., 2009). At sites at higher altitude like Pallas, Finland, or Jungfraujoch, Switzerland, the contribution of charged particle formation was found to be up to 20–

30% (Boulon et al., 2010; Manninen et al., 2010; Kulmala et al., 2013; Rose et al., 2015; Bianchi et al., 2016). In Antarctica, a contribution of 30% was reported (Asmi et al., 2010). From these measurements, it could be inferred that ion-induced nucleation makes only a minor contribution to new particle formation in the boundary layer. However, following ion-induced nucleation, the charged particles are progressively neutralized by recombination with oppositely charged particles. This process, known as ion-ion recombination, needs to be accounted for before the ion-induced nucleation rate can be determined.

The rate at which recombination takes place depends on conditions such as ion concentrations, temperature and humidity (Franchin et al., 2015). The studies estimating the number of recombination-originating neutral clusters using measured ion concentrations have found that very low fractions (0–13%) of clusters formed via recombination compared to total cluster concentrations (Lehtipalo et al., 2009; Manninen et al., 2009; Kontkanen et al., 2013; Kulmala et al., 2013).

The CLOUD experiment measures the ion-induced nucleation rate directly, excluding uncertainties due to subsequent

neutralization of the charged clusters by ion-ion recombination. The method is to compare the nucleation rate measured when high voltage electrodes inside the chamber (Sect. 2.1) are switched on, which rapidly clears out all ions, with the nucleation rate measured with the electrodes set to 0 V (ground potential; Kirkby et al. (2011)). The difference of these two measurements gives the ion-induced nucleation rate due to galactic cosmic rays (GCRs) that traverse the chamber.

The first results from CLOUD indicated that new particle formation involving sulfuric acid, ammonia and water was significantly enhanced by GCR ionization, given that nucleation rates are lower than the limiting ion-pair production rate of about 4 cm$^{-3}$ s$^{-1}$ (Kirkby et al., 2011). In contrast, ion-induced nucleation played only a minor role for particles involving sulfuric acid, dimethylamine and water, due to the high stability (low evaporation rates) of neutral molecular clusters in this case (Almeida et al., 2013; Kürten et al., 2014). A dominant role of ion induced nucleation was found over a wide range of free tropospheric temperatures (249–299 K) for both binary and ternary inorganic particles involving sulfuric acid, ammonia and water (Duplissy et al., 2016). In the case of the recently discovered nucleation of pure biogenic particles, ion-induced nucleation contributed significantly to the total nucleation rate, again up to the limit imposed by the ionization rate (Kirkby et al., 2016).

In this study, we present results on the effect of ions in various atmospherically relevant mixtures of precursor vapors comprising sulfur dioxide (which is oxidized to sulfuric acid), ammonia, monoterpenes (forming highly oxidized molecules, HOMs, Ehn et al. (2014)), NO$_x$ and water, as summarized in Table 1. Furthermore, we were able to determine the contribution of ion-ion recombination to ion-induced new particle formation.

## 2 Methods

### 2.1 Experiment

The CLOUD chamber (Kirkby et al., 2011; Duplissy et al., 2016) is an advanced facility to study nucleation processes, with special emphasis on the control of ions. The temperature-regulated stainless-steel cylinder of 3 m diameter has a volume of 26.1 m$^3$, which provides a wall loss rate comparable to the condensation sink onto aerosol particles in a pristine environment, and long dilution times (2–3 hours, depending on the flow drawn by the sampling instruments). To ensure very low levels of contaminants, all inner surfaces are electropolished and, prior to each experimental campaign, the chamber undergoes a cleaning cycle of several days during which it is first rinsed with ultrapure water and subsequently heated to 373 K while flushing at high rate with humidified ultrapure air containing several ppmv of ozone. Mass spectrometers confirm that the level of contaminants is very low. Concentrations of sulfuric acid and amines are below 10$^5$ cm$^{-3}$. A sophisticated gas supply system is used to control the trace gases added to the chamber when experiments are conducted.

Ions are constantly produced in the chamber by galactic cosmic radiation. Ion concentrations can be further increased by using the CERN Proton Synchrotron (PS) pion beam (3.5 GeV/c) as adjustable additional ionizing radiation. Before the beam traverses the chamber, it is defocused to a transverse size of about 1.5 × 1.5 m. Additional variation in ion concentrations is introduced when aerosol particles in the chamber grow to accumulation mode sizes and act as a sink for small ions. Moreover, 'GCR' ionization rates vary at CLOUD, depending whether the PS is operating or shut down (e.g. for maintenance), since muons from the beam target are able to penetrate the beam stopper. The GCR ionization rate is between 2 i.p. cm$^{-3}$ s$^{-1}$ (PS off) and 4 i.p. cm$^{-3}$ s$^{-1}$ (PS on). During our experiments the PS was mostly operating, however, it was shut down throughout the

measurements for system IV. Ion-free conditions can be studied by using a high-voltage field cage (±30 kV, resulting in 20 kV/m) installed inside the chamber, which efficiently scavenges ions when switched on (ion lifetime below 1 s).

## 2.2 Definitions

The particle nucleation rates reported in this study are defined as follows (details on the calculation are provided in Sect. 2.4). The total nucleation rate is

$$J_{tot} = J_n + J_{iin} \tag{1}$$

where $J_n$ ('neutral') is the nucleation rate in the absence of any ions and $J_{iin}$ is the ion-induced nucleation rate. Previous CLOUD studies (e.g. Kirkby et al., 2016) refer to $J_{tot}$ as $J_{gcr}$ or $J_\pi$ depending on the ionization conditions (solely by galactic cosmic rays or enhanced with the CERN PS $\pi$ beam, respectively). When the nucleated particles are subsequently measured at a larger size, some of the initially-charged particles have been neutralized by ion-ion recombination, so the particle formation rate at a specified detection threshold is

$$J_{tot} = J_n + J_{rec} + J_\pm \tag{2}$$

where $J_{rec}$ is the formation rate of particles that were initially charged, but neutral when detected (ion-ion recombination), and $J_\pm$ is the formation rate of particles, that were initially charged and are still charged when detected. The ion-induced formation rate at the specified detection threshold is

$$J_{iin} = J_{rec} + J_\pm \tag{3}$$

The neutral, i.e. non ion-induced, particle formation rate at the specified threshold is $J_n$ but the *detected* total neutral particle formation rate is

$$J_{n,tot} = J_n + J_{rec} \tag{4}$$

Primary ions in the atmosphere are formed from the most abundant constituents, $N_2$ and $O_2$, which are then positively and negatively charged, respectively. Collisions rapidly transfer the positive charge to vapors with a high proton affinity, such as $H_3O^+$ or $NH_4^+$, and the negative charge to vapors with a high gas phase acidity, such as $CO_3^-$, $NO_3^-$, $HSO_4^-$ (Eisele, 1989; Ehn et al., 2011; Junninen et al., 2010). The ions can attach further molecules such as water. These so-called small ions are singly charged molecules or molecular clusters in the electrical mobility range 3.6–0.6 cm$^2$ s$^{-1}$ V$^{-1}$, corresponding to a mobility diameter 0.75–1.79 nm. Here we refer to small ions as `cluster ions', and their concentration is provided in ion pairs per cubic centimeter (i.p. cm$^{-3}$).

## 2.3 Instruments

A comprehensive set of instruments was used to characterize the chemical and physical properties of the particles and vapors in the CLOUD chamber. Cluster ions and newly formed particles were monitored with ion/particle mobility spectrometers and nano-particle counters. The concentrations and number size distributions of ions were measured with a neutral cluster and air ion spectrometer (NAIS, Airel Ltd.; Mirme and Mirme (2013)). The NAIS simultaneously measures the number size

distribution of positive and negative ions in the mobility diameter range 0.75–45 nm (Mirme and Mirme, 2013) by operating two cylindrical mobility spectrometers in parallel. The sample flow enters the analyzers close to the center-electrode and naturally-charged ions are drifted towards the outer electrodes according to their electrical mobility, transferring their charge onto one of 21 electrometer rings. Taking into account diffusional losses, the spectra of electric currents can be inverted to

number size distributions of ions. After applying calibration corrections, the ion concentrations are accurate to within 10% (Wagner et al., 2016). For operation at CLOUD, a dilution system was employed to reduce the sample flow from the chamber. After including the uncertainty on the dilution correction, the overall uncertainty in ion concentrations is 20%. The NAIS is equipped with sample pre-conditioning units (corona chargers), that can charge the aerosol. In this way neutral aerosol particles can also be measured, in the size range 2–45 nm. The NAIS periodically measures the offset currents of each electrometer by

charging the sample aerosol to the opposite polarity of the subsequent analyzer and switching on an electrical filter. By applying this procedure, no detectable aerosol enters the spectrometers and possible offset currents can be measured and the signals corrected (Manninen et al., 2016).

Particles were measured with two particle size magnifiers (PSM, Airmodus Ltd.; Vanhanen et al. (2011)) together with condensation particle counters (CPC; McMurry (2000)), forming a pair of two-stage nano-particle counters. The PSM operates

with diethylene glycol as the working fluid and achieves supersaturated conditions by mixing heated saturated air with the sample, and subsequently cooling the flow. Since the saturation ratio can quickly be adjusted by altering the flow of saturated air, the cut-off diameter (the diameter with 50% counting efficiency) of the PSM can be varied. In this way, the PSMs were operated in a scanning mode that spanned detection thresholds between approximately 1 and 3 nm. When operated in scanning mode, the number size distributions below 3 nm can be determined (Lehtipalo et al., 2014; Kangasluoma et al., 2015). Particles,

which are activated by the PSM, are subsequently counted by a CPC. In this study, we operated two PSMs in parallel: one of them was measuring all particles, while for another, ions and charged particles were removed from the sample flow with an ion filter. The ion filter consists of two electrodes operated at 2.2 kV potential difference, generating an electric field that removes any ions smaller than approximately 13 nm mobility diameter from the sample flow. The inlet system is described in more detail by Kangasluoma et al. (2016a). The two PSMs, without and with an ion filter, measure the total particle

concentration (PSMt) and the neutral (uncharged) component (PSMn), respectively. The difference between the two PSMs gives the charged fraction. From these particle concentrations ($N_{tot}$, $N_n$) we calculated the formation rates reported in this study. When calculating formation rates, corrections are required for coagulation losses to pre-existing particles. These corrections require knowledge of the particle size distributions, which were measured with two aerosol mobility spectrometers; a nano-SMPS (TSI model 3938; Wang and Flagan (1990)) and a custom-built SMPS. The TSI nano-SMPS was connected to a water

CPC (TSI model 3788) and measured the size distribution in the range 2–65 nm. The custom-built SMPS, consisting of a TSI X-ray source as neutralizer, a TSI-type long differential mobility analyzer (DMA) and a CPC (TSI 3010), measured the size distribution at 20–500 nm. The combination of these two instruments was used to calculate the full size distribution.

The chemical composition of the gases was measured with mass spectrometers and gas monitors. Concentrations of monoterpenes (α-pinene, δ-3-carene) were measured with a proton transfer reaction time of flight mass spectrometer (PTR-

TOF-MS; model: PTR3; Breitenlechner et al., 2017). A new ionization chamber allows for 30-fold longer reaction times and 40-fold higher pressure compared to previous PTR-MS instruments at comparable collision energy. Coupled to the latest quadrupole-interfaced Long-TOF mass analyzer (TOFWERK), sensitivities of up to 20000 cps/ppbv at a mass resolution of 8000 m/$\Delta$m are achieved.

Sulfuric acid and organic HOMs were detected with a chemical ionization atmospheric pressure interface time of flight mass spectrometer (CI-APi-TOF; Jokinen et al., 2012). In this instrument, neutral molecules and clusters are charged by nitrate ions ($NO_3^-$) formed by X-ray ionization of nitric acid in a carrier flow of nitrogen. Nitrate ions then interact with the sample air in an ion drift tube (chemical ionization). After charging, the ions enter the atmospheric pressure interface (APi), where they are focused while the pressure is progressively reduced to $10^{-6}$ mbar. Subsequently the clusters enter the time of flight mass

spectrometer, where their molecular composition is determined by precise mass measurement. Concentrations are subject to a systematic scale uncertainty, as well as uncertainties in charging efficiency in the ion source, a mass dependent transmission efficiency, and sampling line losses (Kirkby et al., 2016). The estimated error of absolute molecule concentrations is roughly a factor of two.

Ammonia ($NH_3$) concentrations were measured with a quadrupole chemical ionization mass spectrometer (CIMS, THS

Instruments LLC). This instrument is equipped with an APi unit (Eisele and Tanner, 1993). Primary ions are formed by ionizing humidified synthetic air with a corona discharge, producing $(H_2O)_n \cdot H_3O^+$ (Kürten et al., 2011). Neutral ammonia molecules in the sample air interact with the ionized water clusters forming $(H_2O)_n NH_4^+$, and are detected mainly as $NH_4^+$. The instrument was calibrated for the relevant range of mixing ratios before and after the experiments by using ammonia from a gas bottle diluted with nitrogen. The limit of detection is approximately 20 pptv of $NH_3$. The error of the measurement was estimated as

a factor of two, which is mainly resulting from the use of different inlet systems during calibration and during operation at the CLOUD chamber.

Nitric oxide (NO) concentrations were determined with a commercial NO monitor (ECO PHYSICS, model: CLD 780 TR) using a chemiluminescence detector. With an integrating time of 60 s, the detection limit is 3 pptv. Nitrogen dioxide ($NO_2$) in the chamber was measured with a cavity attenuated phase shift nitrogen dioxide monitor (CAPS $NO_2$, Aerodyne Research

Inc.). The baseline was monitored periodically by flushing the instrument with synthetic air. Other gas analyzers included the concentrations of sulfur dioxide ($SO_2$, Thermo Fisher Scientific, Inc., model: 42i-TLE), ozone ($O_3$, Thermo Environmental Instruments TEI 49C), and dew-point (EdgeTech).

**2.4 Data analysis**

We present a typical experiment sequence in Fig. 1. The initial conditions were neutral (HV at $\pm$30kV) and so identical

formation rates were measured at 1.5 nm diameter from PSMn, with an electrostatic filter (green curve), and PSMt, without an electrostatic filter (blue curve). Measured ion pair concentrations during that phase of the experiment are solely due to electrometer noise which is scaled up due to corrections for diffusional losses in the sampling line and sample dilution (see Sect. 2.3 for details). When the HV was switched off at 12:02, ions produced by galactic cosmic rays (GCR) were no longer

removed from the chamber and so the concentration of cluster ions increased (Fig. 1c,d). This resulted in an increased particle formation rate due to ion-induced nucleation. As a result of ion-ion recombination, some of the additional ion-induced particles were detected as neutral particles ($J_{rec}$) and the remainder as charged particles ($J_\pm$). In this way, we can measure all four components of the total formation rate: $J_n$, $J_{iin}$, $J_{rec}$ and $J_\pm$. We calculated formation rates at the mobility diameters of 1.5, 2.0, and 2.5 nm, which correspond to mass diameters of about 1.2, 1.7, and 2.2 nm. The size of the smallest detected clusters is similar to HOM di- or trimers, or eight sulfuric acid molecules.

Ion-induced nucleation may depend on numerous parameters, such as chamber temperature, concentration of cluster ions, and concentration of precursors. In this study, we varied these parameters in each studied chemical system to investigate their effect on ion-induced nucleation. A detailed overview of the parameters and corresponding uncertainties is provided in Table 2.

The formation rates ($J$, cm$^{-3}$ s$^{-1}$) were calculated from the time derivative of total particle concentration ($N_{tot}$) above a specified threshold, corrected for the particle loss rates due to dilution, wall losses and coagulation with larger particles (Kirkby et al., 2011; Almeida et al., 2013):

$$J = \frac{dN_{tot}}{dt} + S_{dil} + S_{wall} + S_{coag} \tag{5}$$

Since instruments are continuously sampling from the chamber, a flow of synthetic air is needed to maintain constant pressure. Therefore, the particle concentration in the chamber is diluted at a rate given by

$$S_{dil} = N_{tot} \cdot k_{dil} \tag{6}$$

with $k_{dil} = 1.437 \times 10^{-4}$ s$^{-1}$. Diffusional losses of molecules and particles to the chamber walls ($S_{wall}$) were determined empirically by observing the decay of sulfuric acid monomer concentrations in the chamber after the photochemical production of sulfuric acid was terminated by turning off the UV lights. The wall loss rate is inversely proportional to the mobility diameter of the particle, and can therefore be scaled to determine the wall loss rate for small clusters. Taking into account the dependence on the square root of the diffusion coefficient (Crump and Seinfeld, 1981) and its temperature dependence (Hanson and Eisele, 2000) the wall loss rate can be written as

$$S_{wall}(d_p, T) = \sum_{d'_p=d_p}^{d'_p=max} N(d'_p) \cdot k_{wall}(d'_p, T) \tag{7}$$

$$k_{wall}(d'_p, T) = 2.116 \cdot 10^{-3} s^{-1} \cdot \left(\frac{T}{T_{ref}}\right)^{0.875} \cdot \left(\frac{d_{p,ref}}{d'_p}\right), \tag{8}$$

where $d_p$ is the mobility diameter threshold, $N(d'_p)$ is the concentration of particles with diameter $d'_p$, $d_{p,ref} = 0.82$ nm is the mobility diameter of the sulfuric acid monomer, $T_{ref} = 278$ K, and $T$ is the chamber temperature. The total coagulation loss for particles larger than or equal to $d_{p,k}$ ($S_{coag}(d_{p,k})$) was calculated from the measured number size distribution of particles in the chamber (Seinfeld and Pandis, 2016):

$$S_{coag}(d_{p,k}) = \sum_{d_{p,i}=d_{p,k}}^{d_{p,max}} \sum_{d_{p,j}=d_{p,i}}^{d_{p,max}} \delta_{i,j} \cdot K(d_{p,i}, d_{p,j}) \cdot N_i \cdot N_j \tag{9}$$

with $\delta_{i,j} = 0.5$ if i = j, $\delta_{i,j} = 1$ if i ≠ j, $d_{p,i}$ = midpoint diameter for size bin with index i, $N_i$ = particle number concentration in bin i, and $K(d_{p,i}, d_{p,j})$ = coagulation sink for particles of sizes $d_{p,i}$ and $d_{p,j}$. The nucleation rate for each experimental condition was obtained by calculating the mean of the nucleation rates measured after reaching stable conditions. To ensure a high quality data set, we discarded results where the relative standard deviation of the nucleation rate was larger than 0.3.

When studying the ratio of total to neutral nucleation rates, we compared measurements from two PSMs. In general, the agreement of the two instruments during neutral conditions was very good. However, the few cases (<1% of all measurements), where the formation rate of neutral particles ($J_{n,tot}$) exceeded the formation rate of total particles ($J_{tot}$) by more than 30%, were excluded from the analysis. This sometimes occurred due to measurement uncertainties when nucleation rates were very low (<$10^{-3}$ cm$^{-3}$ s$^{-1}$).

Uncertainties in the ratios of total to neutral nucleation rates were calculated from the uncertainties of the concentration measurements, as well as the sink terms. Beyond that, there are a few more limitations to our method.

One source of uncertainty is the composition dependency of the detection thresholds of the PSMs. The instruments were calibrated using tungsten oxide particles before the measurement campaign. However, a higher detection threshold has been

reported for organic particles (Kangasluoma et al., 2014). To account for this we compared the cut-off diameters of the PSM to the size bins of the NAIS in each chemical system used here, and chose the diameters based on this comparison. The NAIS is insensitive to composition as it detects the size based on ion mobility, and the size accuracy has been verified in laboratory calibrations (Wagner et al., 2016). The remaining uncertainty is in the order of ± 0.2 nm based on limited size bin resolution and run-to-run variability.

When comparing the PSMt and PSMn measurements, a charge effect on the instruments' detection efficiency might further affect our results. Ions are known to activate at lower supersaturations compared with neutral particles (Winkler et al., 2008). For the PSM, the cut-off diameter for ions can be up to 0.5 nm smaller than for neutral particles, depending on particle composition (Kangasluoma et al., 2016b). In practice, the detected ions could be a bit smaller than the neutral particles at the same saturation ratio. As a result, depending on the particle growth rate, the ratio $J_{\pm}/J_{tot}$ would be slightly increased (the ratio

$J_{n,tot}/J_{tot}$ slightly decreased). Although we do not expect this charge effect to be significant in our study, we want to point out that the reported charged fractions represent upper-limit estimates.

Further quantification of the effect of charge and composition on the detection threshold would require extensive knowledge on the particle and cluster composition and their activation properties in each system, and is left for future studies.

**3 Results**

**3.1 Fraction of neutral particle formation in different chemical systems**

We will use the term 'neutral fraction' at a given detection threshold to indicate the measured ratio of the neutral to total formation rates, $J_{n,tot}/J_{tot}$. Figure 2 illustrates the neutral fraction of all four systems combined, at several detection size thresholds. A progressive neutralization of the clusters can be seen as the particles grow in size; the median neutral fractions are 0.54, 0.72 and 0.95 at 1.5 nm, 2.0 nm and 2.5 nm threshold, respectively. While an exponential decrease of the charged fraction was reported in an earlier study (Yu and Turco, 2011), we observed a linear decay. However, the charging state is sensitive to the age of the sample, which may be different in our study (characteristic mixing time; see Sect. 3.2) compared to the data analyzed by Yu and Turco (2011).

The first chemical system we studied (system I) contained biogenic vapors alone. Monoterpenes (MT; α-pinene, δ-3-carene, or a mixture, $C_{10}H_{16}$) injected into the chamber were subsequently oxidized by ozone and hydroxyl radical (OH), forming HOMs. We found that the importance of charge decreased towards high MT concentrations (Fig. 3a). Although we study the neutral fraction here, which includes neutral nucleation and recombination of ion-induced particles, the observed behavior indicates that ion-induced nucleation also follows this pattern. This was previously reported by Kirkby et al. (2016), as a result of $J_{iin}$ saturating at the GCR ion production rate limit. At low temperatures, all HOM species have reduced volatility and so a larger fraction can participate in particle nucleation and growth - although this is partially compensated by the slower production rate of HOMs. Temperature also affects the composition and stability of formed HOMs clusters (Frege et al., 2017). As a result, the neutral fraction at a given MT concentration is higher at lower temperatures (Figs. 3a and 3b). Compared to 1.5 nm, particles reaching 2.0 nm in diameter had more time to get neutralized by ion-ion recombination, and were already more stable so the charge was less important to stabilize them (Fig. 3b). Particles measured at 2.5 nm detection threshold were mostly neutral at all studied conditions (Fig. 3c).

With the addition of sulfur dioxide (system II) the influence of charge depended on the concentrations of both monoterpenes and sulfuric acid. We therefore studied the neutral fraction as a function of the product of the concentrations of monoterpenes and sulfuric acid (Fig. 4), since only then the trends became clearly visible. The observed decrease of the charged fraction at the lowest temperature (Fig. 4a, compared to Fig. 3a) suggests higher cluster stability when sulfuric acid is present. Otherwise we observed trends similar to system I. Once again, particles measured at 2.5 nm detection threshold were mostly neutral at all studied conditions (Fig. 4c).

After addition of NO (system III) to study the possible effect of $NO_x$ on new-particle formation, the gas mixture comprised monoterpenes, sulfuric acid and nitrogen oxides (NO and $NO_2$). $NO_x$ are found to decrease the particle formation rates from monoterpene oxidation in previous studies (Wildt et al., 2014). Here, we found a decreasing neutral fraction with increasing

concentrations of NO and of cluster ions. We therefore show in Fig. 5 the neutral fraction versus $[MT]*[H_2SO_4]/([NO]*[cluster ions])$. The neutral fraction decreased towards lower values of this quantity (Fig. 5a). For this system, the nucleation rate is primarily driven by HOMs rather than sulfuric acid, so a repeated pattern can be seen at various $[H_2SO_4]$ levels in Fig. 5a. However, sulfuric acid adds to the stability of 2.0 nm particles, as the neutral fraction is lowest with $[H_2SO_4]$ below the detection limit of $10^5$ cm$^{-3}$. As before, particles measured at 2.5 nm detection threshold were mostly neutral at all studied conditions (Fig. 5c).

With the addition of ammonia we aimed to reproduce an environment similar to the boreal forest at the station for measuring ecosystem-atmosphere relations (SMEAR II, Hari and Kulmala (2005)) in Hyytiälä, southern Finland, involving a mixture of monoterpenes, sulfuric acid, nitrogen oxides, and ammonia (system IV). During new particle formation events, typical conditions in Hyytiälä are [cluster ions] = 440–580 i.p. cm$^{-3}$, [MT] = 30–140 pptv, $[H_2SO_4]$ = 4–8 × 10$^6$ cm$^{-3}$, [NO] = 20–90 pptv, [NO$_2$] = 260–1130 pptv, [NH$_3$] = 50–210 pptv, and T = 3–14 °C. The values in the ranges correspond to the 25$^{th}$ and 75$^{th}$ percentiles. The dependency of the neutral fraction on the different variables in this system seemed to be similar to system III, although the neutral fractions especially at 1.5 nm were clearly higher. The neutral fraction of particle formation rates at 1.5 nm ranged from about 10% at the low MT and $H_2SO_4$ concentrations up to 80–90% at the high concentrations (Fig. 6a). The latter corresponded to T ≈ 5 °C, [MT] ≈ 690 pptv, $[H_2SO_4]$ ≈ 10$^7$ cm$^{-3}$, NH$_3$ ≈ 180 pptv, [NO] ≈ 20 pptv and [cluster ions] ≈ 600 i.p. cm$^{-3}$ and, under these conditions, $J_n$ exceeds the ion production rate limit for $J_{iin}$. In this multi-component system, ammonia helps to stabilize the sulfuric acid so the neutral fraction of particle formation at 1.5 nm and 5°C (Fig. 6a) is larger towards lower MT and $H_2SO_4$ concentrations than seen in Fig. 5a (for $H_2SO_4$ > 3 × 10$^6$ cm$^{-3}$). We speculate that this is due to a similar base-stabilization mechanism, as observed in Kirkby et al. (2011) for a ternary sulfuric acid-water-ammonia system, although the multi-component system studied here is more complicated than pure acid-base systems. Ions are still important in stabilizing the particles at warmer temperatures (Fig. 6a, 25°C). As for all other systems, particles measured at 2.5 nm detection threshold were mostly neutral at all studied conditions (Fig. 6c).

We display a comparison of the neutral fractions of particle formation rates at 5°C for all four systems in Fig. 7. Examining the smallest studied clusters (1.5 nm, Fig. 7a) demonstrates the significance of ions for all systems, and also that ammonia helps stabilizing the clusters, reducing the importance of the charge. As the particles grow, charged particles are gradually neutralized by ion-ion recombination (Fig. 7b) until reaching 2.5 nm, when less than 10% of all particles carry a charge (Fig. 7c). Values larger than one result from nucleation rates close to detection limit (approximately $10^{-3}$ cm$^{-3}$ s$^{-1}$).

**3.2 Comparison of CLOUD measurements to atmospheric observations at SMEAR II, Hyytiälä, Finland**

We compare in Fig. 8 and 9 the CLOUD nucleation and formation rates with those reported from several atmospheric studies conducted in Hyytiälä. We compared the 1.5 nm formation rates in CLOUD with the nucleation rates of 1.5 nm particles (Kulmala et al., 2013), and the recombination rates of 1.5–1.7 nm particles (Kontkanen et al., 2013). In addition, we compared

the formation rates of 2.0 nm particles in CLOUD with the formation rates at 2 nm from Manninen et al. (2009). Most of the Hyytiälä measurements were carried out in spring, when the temperatures ranged between around -5 to 15°C (median 6.3°C).

In Fig. 8, we compare CLOUD (system IV) and Hyytiälä measurements of the neutral and ion-induced nucleation rates versus

cluster ion concentrations. At CLOUD, the fractions of pure neutral and ion-induced particle formation do not depend on the particle detection threshold. That means, although the total particle formation rate decreases with increasing detection threshold diameter, the relative contribution of ion-induced nucleation remains the same. The $J_{iin}/J_{tot}$ fraction increases with cluster ion concentration from about 25% at the lowest ion concentrations, 580 i.p. cm$^{-3}$, to more than 90% at 1230 i.p. cm$^{-3}$ (Fig. 8d). The ion-induced fraction in Hyytiälä at 1.5 nm (triangle, Fig. 8d) is almost one order of magnitude below the values at CLOUD,

but the cluster ion concentration is also respectively lower than those in CLOUD. From Fig. 6 it is clear that the neutral and ion-induced fractions depend on the cluster ion concentration in this chemical system. The difference is smaller at 2.0 nm detection threshold, however, the atmospheric values are still roughly a factor of two lower than at CLOUD (triangle and diamond, Fig. 8e).

For comparison, we display in Fig. 9 the measured recombination and charged fractions of the particle formation rates versus cluster ion concentrations for CLOUD (system IV) and Hyytiälä. Comparison of ion-induced and charged fractions at CLOUD at 1.5 nm threshold (Figs. 8d and 9d) show that a fraction of the ion-induced particles has already been neutralized by ion-ion recombination, even at 1.5 nm detection threshold. This follows since the mean age of the particles sampled by the PSMn or PSMt at any instant in time includes the characteristic mixing time in the CLOUD chamber, which is several minutes and

comparable to the ion-ion recombination lifetime. Consequently, even at the lowest threshold of 1.5 nm, a fraction of the ion-induced charged particles have been neutralized before they are sampled by the PSMn and PSMt. Therefore the charged fraction $J_{\pm}/J_{tot}$, even when measured at 1.5 nm, cannot be simply interpreted as the ion-induced fraction. Figure 9d–f illustrates once more that the charged particles are progressively neutralized by ion-ion recombination until reaching 2.5 nm, when they represent less than 10% of the total (Fig. 9f). The Hyytiälä charged and recombination fractions at the lowest particle size,

1.5 nm, are approximately one order of magnitude below the CLOUD measurements. Again, differences are smaller at 2.0 nm. However, with the contributions of charged particle formation and recombination, a comparison between CLOUD and Hyytiälä is more difficult, since those quantities depend not only on the initial conditions, but on ion-aerosol dynamics. Key parameters are nuclei growth rate and concentration of cluster ions (Kerminen et al., 2007).

An important difference between CLOUD and Hyytiälä is that the ion concentrations measured in CLOUD under GCR conditions are higher than those measured at Hyytiälä. At CLOUD, the ion pair concentrations are about 1000 cm$^{-3}$ (total concentration of cluster ions around 2000 cm$^{-3}$) when the CERN PS is off i.e. when no beam pions or muons traverse the chamber. Together with the measured wall loss rate for ions of $1.4 \times 10^{-3}$ s$^{-1}$, the ion concentrations measured in CLOUD are within the expected range, with the known GCR ion pair production rate of 2 cm$^{-3}$ s$^{-1}$. On the other hand, at Hyytiälä, the mean

ion pair concentrations are about one half of the CLOUD value, around 500 cm$^{-3}$, whereas the maximum ion pair production rate (ionizing capacity) is 6–12 cm$^{-3}$ s$^{-1}$, which is 3–6 times higher than the ion pair production rate at CLOUD, due to the additional ionization contribution from radon decay (Chen et al., 2016). The wall loss rate in CLOUD is comparable to the condensation sink in the atmosphere under pristine conditions. In Hyytiälä, on average, a condensation sink of $2.5 \times 10^{-3}$ s$^{-1}$ is

observed (Nieminen et al., 2014). However, the cluster ion concentrations at Hyytiälä are found to be relatively insensitive to condensation sink (see, for example, Fig. 12 in Chen et al., 2016), so the discrepancy in ion concentrations between Hyytiälä and CLOUD cannot be explained as a difference in average condensation sink. Rather a great variety of other sinks, such as manifold surfaces like canopy or ground, or the atmospheric electrical field repelling negative ions from the surface (Tammet et al., 2006), is most probably responsible for the lower concentrations of cluster ions in Hyytiälä. The sum of these additional

sinks appears to be of the magnitude $5 \times 10^{-3}$ s$^{-1}$ (CS $\approx 8 \times 10^{-3}$ s$^{-1}$), since with this additional sink present, cluster ion concentrations in CLOUD are comparable to Hyytiälä (Fig. 10).

## 4 Conclusions

We have used a novel instrument setup at CLOUD comprising two nano-particle counters, one of them equipped with an ion filter, to measure the charged and neutral particle concentrations at detection thresholds between 1.5 nm and 2.5 nm, for several

different mixtures of precursor vapors. We have compared the neutral and ion-induced nucleation rates measured at CLOUD with the corresponding nucleation rates measured at the SMEAR II station in Hyytiälä, Finland.

We find that charged clusters are efficiently neutralized by ion-ion recombination. While in some cases around 90% of 1.5 nm clusters were charged, just roughly 10% still carried a charge when they grew to 2.5 nm. CLOUD measurements of the ion-

induced nucleation rate, $J_{iin}$, are unaffected by ion-ion recombination since they are obtained from measurements made with the high voltage clearing field switched on (measuring $J_n$ alone) and off (measuring $J_n + J_{iin}$). On the other hand, measurements of $J_{iin}$ in field experiments require correction for recombination losses since they rely on the detection of charged clusters.

Our results indicate that ions significantly enhance the nucleation rates in almost all the chemical systems that have been

studied so far in the CLOUD chamber - provided the nucleation rate does not exceed the ionization rate limit. The notable exception is H$_2$SO$_4$-dimethylamine, which forms highly stable neutral particles at near the H$_2$SO$_4$ kinetic limit (Kürten et al., 2014) and so, for this system, ions add insignificant additional cluster stability. When simulating the conditions in a boreal forest (system IV), we find that ion-induced nucleation contributes to the total nucleation rate between 25% at cluster ion concentrations comparable to Hyytiälä, and 90% at cluster ion concentrations roughly a factor of two higher than in Hyytiälä.

Measurements at Hyytiälä find that ion-induced nucleation accounts for around 9–15% of total new particle formation (Manninen et al., 2009). An important difference between CLOUD and Hyytiälä is that the ion concentrations measured in CLOUD under GCR conditions are higher than those measured at Hyytiälä, even though the ion pair production rate under

GCR (zero beam) conditions is a factor 3–6 lower at CLOUD than at Hyytiälä. The origin of the discrepancy in ion concentration between the CLOUD laboratory measurements and the Hyytiälä field measurements is not yet known in detail and indicates the need for further investigation.

**Data availability**

Data that was used to create the presented tables and figures can be downloaded from Zenodo at DOI 10.5281/zenodo.1033853.

**Acknowledgements**

We thank the European Organization for Nuclear Research (CERN) for supporting CLOUD with important technical and financial resources and providing a particle beam from the CERN Proton Synchrotron. We also thank P. Carrie, L.-P. De Menezes, J. Dumollard, K. Ivanova, F. Josa, I. Krasin, R. Kristic, A. Laassiri, O. S. Maksumov, B. Marichy, H. Martinati, S.

V. Mizin, R. Sitals, A. Wasem and M. Wilhelmsson for their contributions to the experiment. This research was supported by the EC Seventh Framework Programme (Marie Curie ITN 'CLOUD-TRAIN', no. 316662; Advanced Grant, 'ATMNUCLE', no. 227463, Consolidator Grant 'NANODYNAMITE', no. 616075; Starting Grant 'QAPPA', no. 335478; Starting grant 'MOCAPAF', no. 257360; MC-COFUND 'CERN-COFUND-2012', no. 600377), EC Horizon 2020 (Marie Skłodowska-Curie Individual Fellowships 'Nano-CAVa', no. 656994; Starting Grant 'COALA', no. 638703), German Federal Ministry of

Education and Research (nos. 01LK1222A and 01LK0902A), Presidium of the Russian Academy of Sciences and Russian Foundation for Basic Research (grants 08-02-91006-CERN and 12-02-91522-CERN), U.S. Department of Energy (no. DE-SC0014469), Swiss National Science Foundation (206620_141278, 200020_135307, 200021_140663, 206021_144947/1), Austrian Science Fund (J3198-N21, L593, P19546), Portuguese Foundation for Science and Technology (CERN/FP/116387/2010), Swedish Research Council (2011-5120), U.S. National Science Foundation (AGS1136479,

AGS1447056, AGC1439551, CHE1012293), Natural Environment Research Council (NE/J024252/1, NE/K015966/1), Royal Society (Wolfson Merit Award), Dreyfus Award (EP-11-117), European Funds for Regional Economic Development (Labex-Cappa Grant, ANR-11-LABX-0005-01), Nessling Foundation, Finnish Funding Agency for Technology and Innovation, Väisälä Foundation, Caltech Environmental Science and Engineering Grant (Davidow Foundation), French National Research Agency, Nord-Pas de Calais, and the Academy of Finland (nos. 299574, 1118615, 135054, 133872, 251427, 139656, 139995,

25  141451, 137749, 141217, 138951).

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

**Table 1.** Overview of the four precursor vapor mixtures investigated in the present study. The precursors were added to the chamber at various atmospheric concentrations together with 40 ppbv (parts per billion by volume) ozone and ultra-pure synthetic air ($N_2/O_2 = 79/21$) at 38% relative humidity.

| System no. | I | II | III | IV |
|---|---|---|---|---|
| Monoterpenes (MT) | ✓ | ✓ | ✓ | ✓ |
| Sulfur dioxide ($SO_2$) | | ✓ | ✓ | ✓ |
| Nitric oxide (NO) | | | ✓ | ✓ |
| Ammonia ($NH_3$) | | | | ✓ |

5    **Table 2.** Experimental ranges of temperatures (T), CERN proton synchrotron (PS) beam intensities, total ion pair production rates (IPR), and concentrations of cluster ions (mobility diameter 0.75–1.8 nm), monoterpenes (MT), biogenic highly oxidized molecules (HOM), sulfuric acid ($H_2SO_4$), nitric oxide (NO), nitrogen dioxide ($NO_2$), and ammonia ($NH_3$), and the corresponding uncertainties.

| | System I | | System II | | System III | | System IV | | |
|---|---|---|---|---|---|---|---|---|---|
| | min | max | min | max | min | max | min | max | Uncertainty |
| T (°C) | -25.2 | 25.5 | -25.2 | 25.5 | 5.0 | 5.3 | 5.2 | 25.4 | ±0.1 |
| PS beam intensity [Hz] | <3E+03 | 8.0E+04 | <3E+03 | 5.1E+04 | <3E+03 | <3E+03 | n/a | n/a | ±10% |
| Total IPR [i.p. cm$^{-3}$ s$^{-1}$] | 4.4 | 54.9 | 4.4 | 35.7 | 4.4 | 4.4 | 1.8 | 1.8 | ±20% |
| [Cluster Ions] (i.p. cm$^{-3}$) | 1.0E+03 | 5.8E+03 | 9.2E+02 | 5.6E+03 | 1.2E+02 | 2.9E+03 | 6.1E+02 | 1.2E+03 | ±20% |
| [MT] (pptv) | 98 | 1956 | 28 | 1540 | 253 | 1578 | 134 | 1397 | ±15% |
| [HOM] (cm$^{-3}$) | 1.1E+06 | 3.8E+07 | <1E+06 | 2.4E+07 | 6.2E+06 | 3.5E+07 | <1E+06 | 1.8E+07 | +100%/−50% |
| [$H_2SO_4$] (cm$^{-3}$) | <1E+05 | <1E+05 | 1.1E+06 | 1.0E+08 | <1E+05 | 2.3E+07 | 1.6E+06 | 7.3E+07 | +100%/−50% |
| [NO] (ppbv) | 0.002 | 0.019 | 0.001 | 0.012 | 0.005 | 0.084 | 0.015 | 0.033 | ±0.020 |
| [$NO_2$] (ppbv) | n/a | n/a | n/a | n/a | 0.038 | 13.499 | 0.052 | 2.065 | ±0.200 |
| [$NH_3$] (pptv) | n/a | n/a | n/a | n/a | n/a | n/a | 178 | 1971 | ±35% |

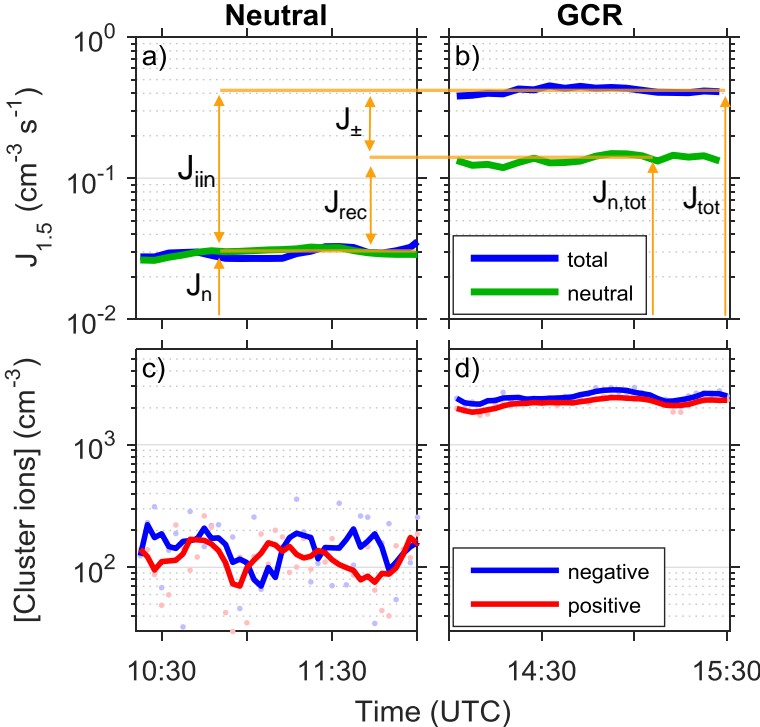

**Figure 1. Example of an experimental run to illustrate individual particle formation rates, measured at 1.5 nm detection threshold. Panels a and b: particle formation rates measured by the PSMt (without ion filter, blue curve) and PSMn (with ion filter, green curve); c and d: cluster ion concentrations ($d_p$ 0.75–1.8 nm) measured by the NAIS. Prior to 12:02 UTC the high voltage (HV) clearing field was on to establish ion-free conditions in the chamber and so PSMt and PSMn measured the same formation rates ($J_n$). After switching off the HV, the ions produced by galactic cosmic rays (GCR) were no longer removed from the chamber (panel d) and the particle formation rates increased (panel b). The increase in particle formation rate measured by PSMt provides the ion-induced formation rate ($J_{iin}$), and the increase in particle formation rate measured by PSMn provides the fraction of $J_{iin}$ that is detected as neutral particles, due to ion-ion recombination. The difference of PSMt and PSMn signals provides the fraction of $J_{iin}$ detected as charged particles. The run conditions are T = 5.2 °C, [MT] = 270 pptv, [O$_3$] = 40 ppbv; [H$_2$SO$_4$] = 1.4 × 10$^7$ cm$^{-3}$, [NO] = 0.084 ppbv (system III).**

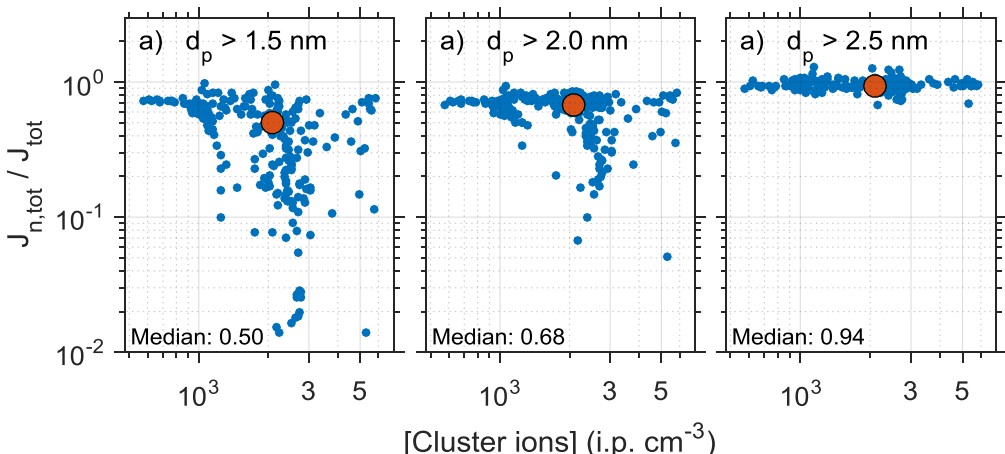

**Figure 2. The neutral fraction of particle formation rates measured at detection thresholds of a) 1.5 nm, b) 2.0 nm, and c) 2.5 nm, versus cluster ion concentrations. All four systems are included. Each red dot indicates the median neutral fraction and ion pair concentration. Whereas ion-induced nucleation can result in large charged fractions at the smallest detection threshold, 1.5 nm (panel a), more than 90% of particles are neutral once they reach 2.5 nm (panel c).**

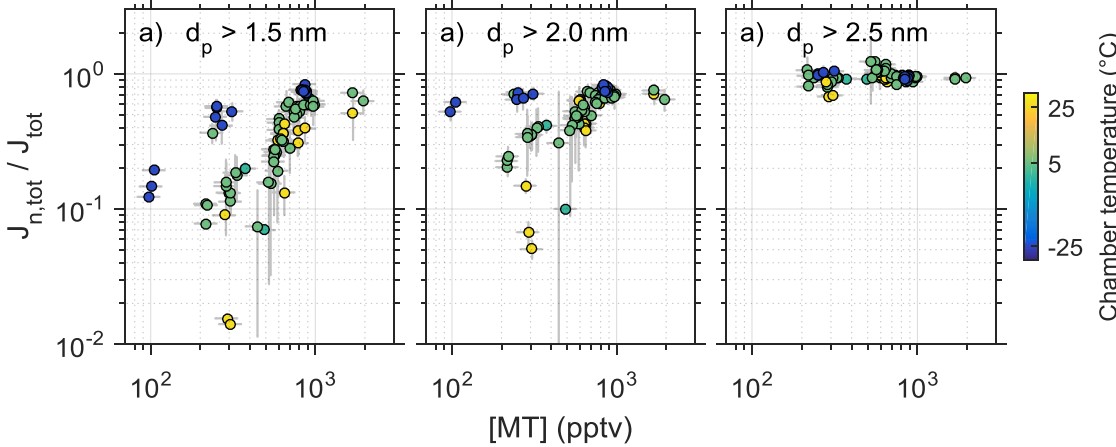

**Figure 3. The neutral fraction of particle formation rates versus monoterpene (MT) concentration for pure biogenic conditions (system I), at detection thresholds of a) 1.5 nm, b) 2.0 nm, and c) 2.5 nm. The color scale indicates chamber temperature (-25°C, 5°C, 25°C).**

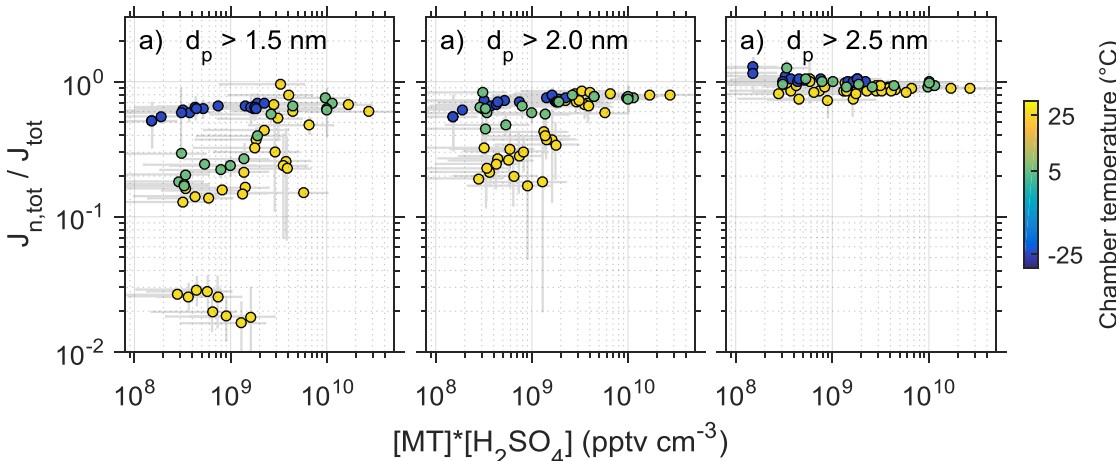

**Figure 4. The neutral fraction of particle formation rates versus the product of the monoterpene and sulfuric acid concentrations (system II), at detection thresholds of a) 1.5 nm, b) 2.0 nm, and c) 2.5 nm.**

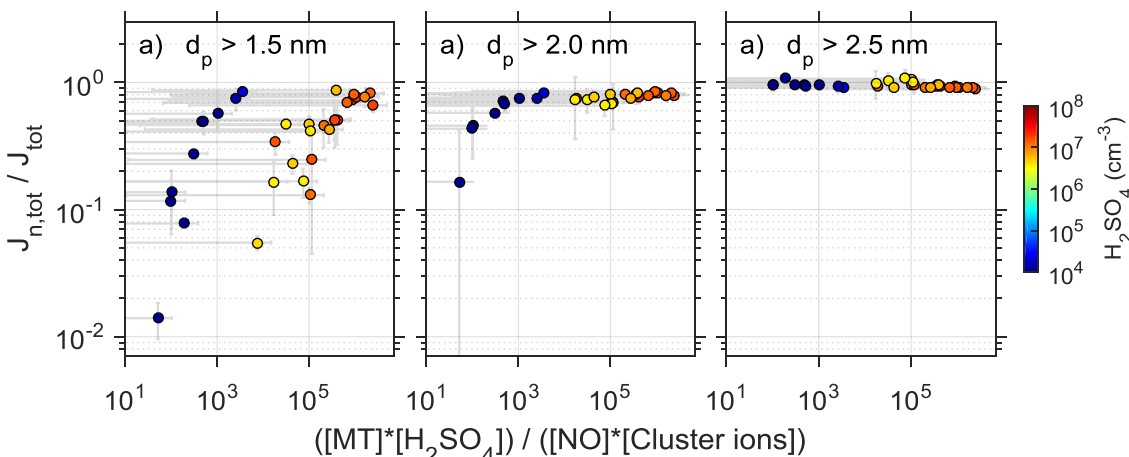

**Figure 5. The neutral fraction of particle formation rates versus the product of the concentrations of monoterpenes (MT) and sulfuric acid (H2SO4) divided by the concentration of nitric oxide (NO) and cluster ions (system III), at 5°C temperature and detection thresholds of a) 1.5 nm, b) 2.0 nm, and c) 2.5 nm.**

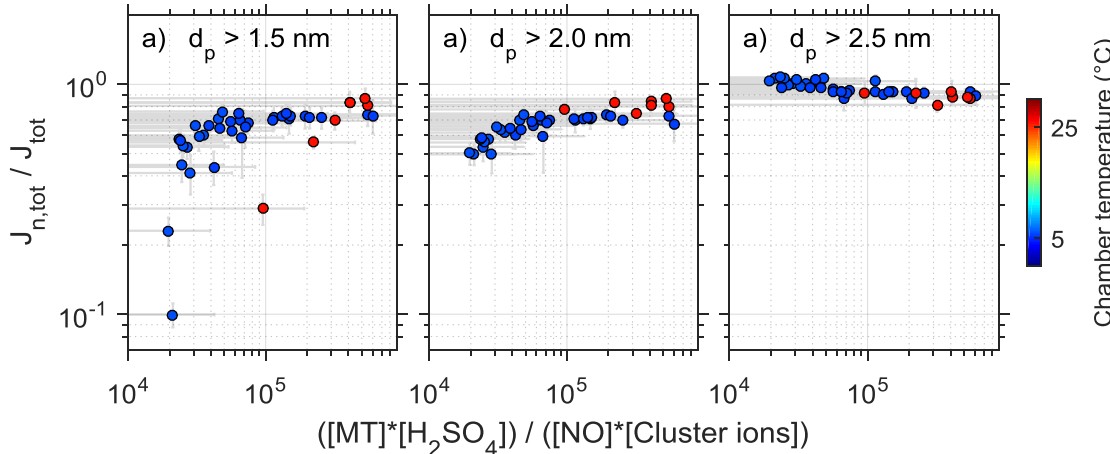

**Figure 6.** The neutral fraction of particle formation rates versus the product of the concentrations of sulfuric acid and monoterpenes divided by the concentrations of nitrogen oxide (NO) and cluster ions, after adding ammonia (NH3) to the chamber (Hyytiälä simulation, system IV), at detection thresholds of a) 1.5 nm, b) 2.0 nm, and c) 2.5 nm.

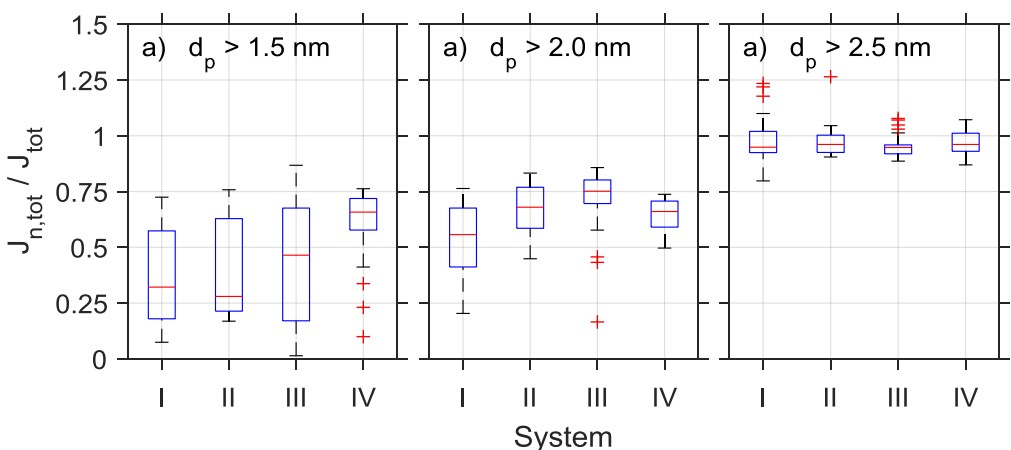

**Figure 7.** Comparison of the neutral fraction of particle formation rates for all four chemical systems at 5°C and detection thresholds of a) 1.5 nm, b) 2.0 nm, and c) 2.5 nm. The box and whisker plots show the median (red line), upper and low quartiles
10 (rectangular box) and upper and lower range (error bars). Red crosses indicate outliers.

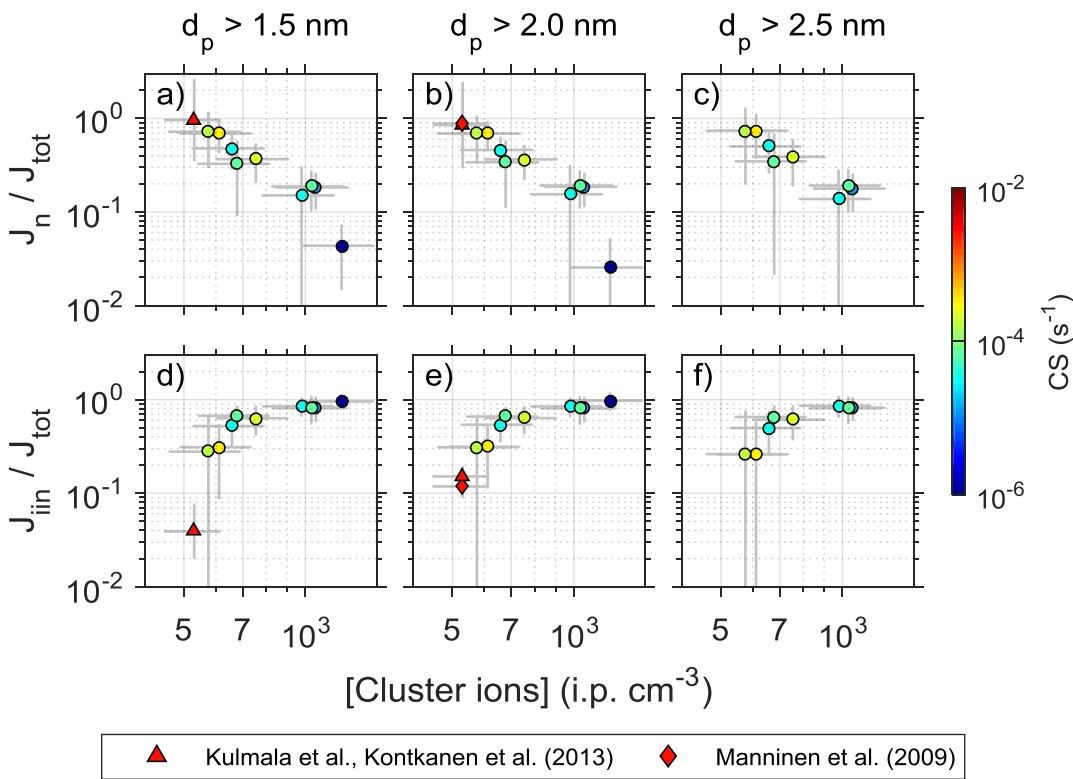

**Figure 8. Comparison of CLOUD (system IV; circles) and Hyytiälä, Finland, (triangles and diamonds) measurements of the neutral and ion-induced fractions of particle nucleation rates versus cluster ion concentrations at 5 and 25°C and detection thresholds of a,d) 1.5 nm, b,e) 2.0 nm, and c,f) 2.5 nm. The color scale indicates the condensation sink (CS) onto aerosol particles (wall loss and dilution loss not included). The condensation sink in Hyytiälä is on average $2.5 \times 10^{-3}$ cm$^{-3}$ (Nieminen et al., 2014).**

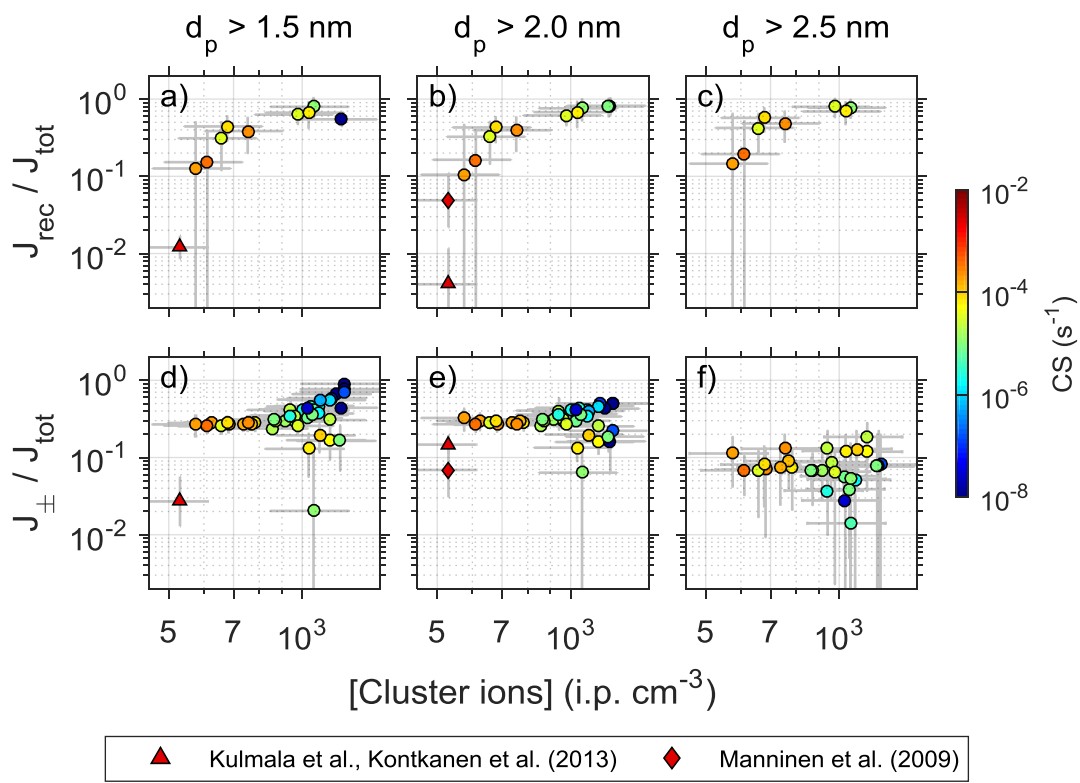

**Figure 9. Comparison of CLOUD (system IV; circles) and Hyytiälä, Finland, (triangles and diamonds) measurements of the charged and recombination fractions of particle formation rates versus cluster ion concentrations at 5°C and detection thresholds of a,d) 1.5 nm, b,e) 2.0 nm, and c,f) 2.5 nm. The color scale indicates the condensation sink (CS) onto aerosol particles (wall loss and dilution loss not included). The condensation sink in Hyytiälä is on average $2.5 \times 10^{-3}$ cm$^{-3}$ (Nieminen et al., 2014).**

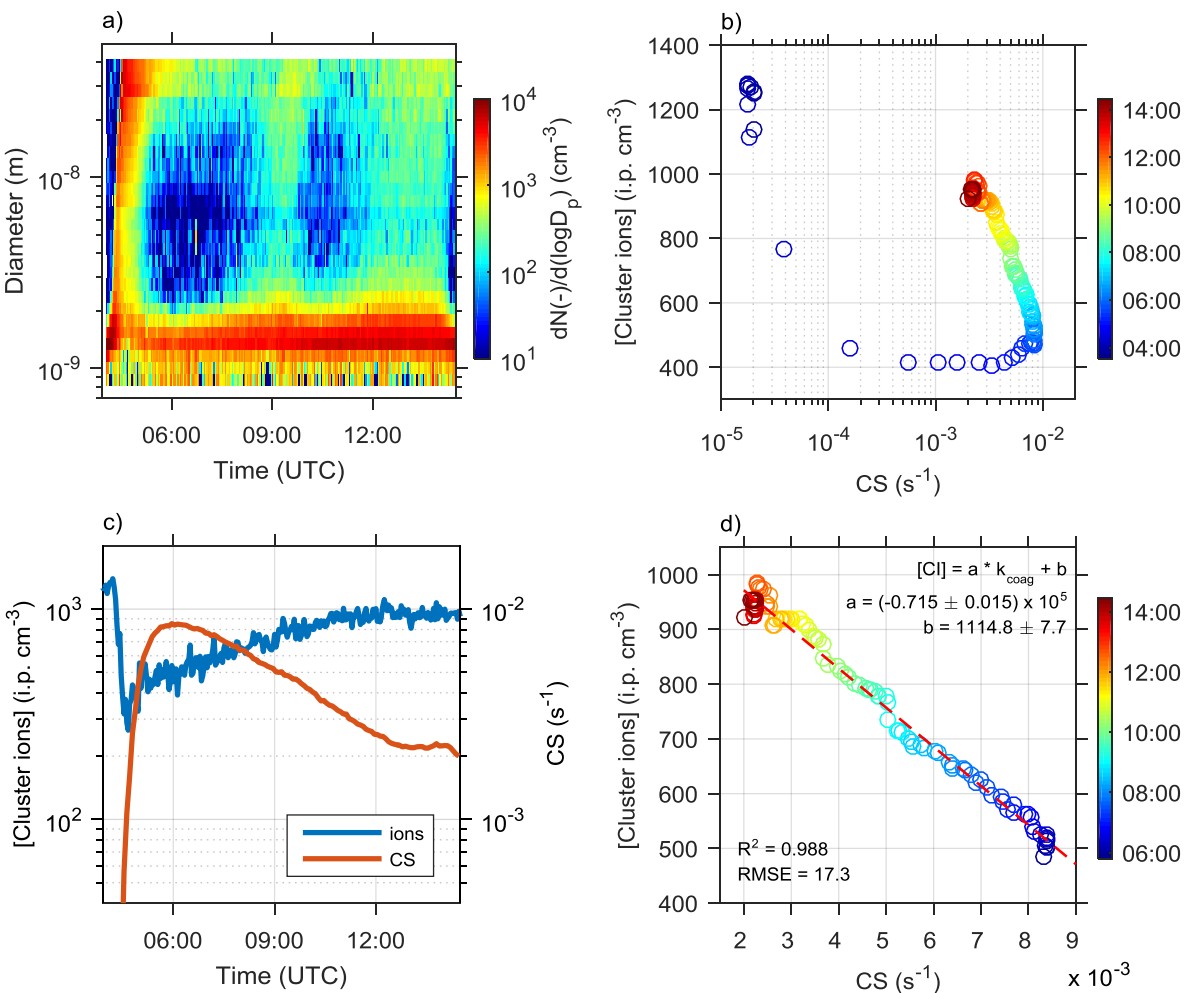

**Figure 10. Evolution of the concentration of cluster ions and condensation sink (CS) during an experiment with high nucleation and growth rate ($J_{tot} \approx 8$ cm$^{-3}$ s$^{-1}$, GR $\approx 80$ nm h$^{-1}$). The rapid ion-induced nucleation burst at around 4:00 UTC (N(-), panel a)**
5 **partially depletes the pool of cluster ions (panel c). At 6:00 the particle concentration in the chamber is 6200 cm$^{-3}$, with a mode diameter near 150 nm, and the condensation sink, CS, is $8.5 \times 10^{-3}$ s$^{-1}$ (panel c). The large condensation sink quenches nucleation of further particles, due to vapor depletion. The particles are then diluted out of the chamber over a period of around 6 hours, during which time the condensation sink falls from $8.5 \times 10^{-3}$ s$^{-1}$ to $2 \times 10^{-3}$ s$^{-1}$ (panels b and d). A tight correlation is observed between condensation sink and concentration of cluster ions ($d_p$ 0.75–1.8 nm), illustrating the direct influence of the aerosol particle**
10 **condensation sink on ambient cluster ion concentrations (panel d).**

