# Peer review of "The role of ions in new-particle formation in the CLOUD chamber"

_Atmospheric Chemistry and Physics, 2017_

## Referee Comment (RC1) · Anonymous Referee #1 · 3 Aug 2017

Ion-induced nucleation has been widely accepted as an important source of new-particle formation, as well as a famous argument for its ratio in atmosphere. This manuscript aims to reveal the ion role via a well-designed experiment at the CERN CLOUD chamber with novel characteristic instruments. This study presents very important experimental data to support the enhancement of ions in the nucleation process. The first time the ion contribution has been examined in such detail. In general, this manuscript was well-organized and the main conclusions will help improve the current understanding of new-particle formation. This manuscript should be published in ACP. I suggest a little more discussion and analysis to clarify the details behind the presented results. Specific comments:

1. Page 9, line 30: "...a linear decay." If the charged fraction was a linear decay in the

averaged experimental data, the author possibly could further estimate the charged fraction in a lower diameter, such as < 0.5 nm, which can reveal the role of ions in the initial stage of nucleation.

2. Page 10, line 7-10: the authors should give more detail discussion to explain the decrease of charged fraction of nucleation rate at higher temperature (figure 3 a).

3. Page 10, line 10-12: since the ion-ion recombination increases at a larger size (such as 2.2 nm), the authors possibly can make estimation to get the ion-induced fraction at the initial nucleation stage at a molecular level. The ion contribution of nucleation at molecular level will be toward the final answer for the role of ion-induced nucleation.

4. Page 10, line 21-25: the author should give more detail description on the characteristics and roles of NOx in the nucleation process to distinguish the system II and III.

5. Page 10 line 29 – page 11 line 8, last paragraph: ammonia was added in the system IV to reproduce an environmental simulation. Since the ammonia ion is easier to carry positive charges, its role in the nucleation was described as a help to stabilize the sulfuric acid. I think the authors possibly could present a more detail explanation as the ion-ion recombination.

---

## Referee Comment (RC2) · Anonymous Referee #2 · 2 Oct 2017

The authors reported results from experiments at CLOUD 5 on four systems of different chemical compositions involving monoterpenes, sulfuric acid, nitrogen oxides, and ammonia. With instrument setup consisting of two nano-particle counters, one of them equipped with an ion filter, the authors were able to investigate the effect of ions on nucleation and measure the progressive neutralization due to ion-ion recombination as clusters grow. The measurements indicate that ions enhance the nucleation process when the charge is necessary to stabilize newly formed clusters, and a large fraction of the clusters carried a charge at 1.2 nm diameter but most of these charged clusters were largely neutralized before they grew to 2.2 nm. The authors also compared CLOUD measurements to atmospheric observations at SMEAR II, Hyytiälä, Finland.

The measurements and data analysis presented in this manuscript are important to

better understand the role of ions in new particle formation under different conditions. The manuscript is well within the scope of ACP. Some clarifications and additional details are needed to further improve the manuscript. I recommend the publication of this manuscript in ACP after the following comments are addressed.

Main comments

1. Figures 3-6. These figures present neutral fraction of particle formation rates versus [MT] or [MT] combined with other parameters ([H2SO4], [NO], [Cluster ions]). It is established that HOMs (from the oxidation of MT) are involved in the nucleation and/or growth of clusters. Since [HOMs] are measured (page 7) and it appears that [HOMs] are affected by other parameters such as temperature (lines 7-9, page 10), I think that the results will be more straightforward and easier to understand if [MT] in x-axis is replaced with measured [HOMs] and the figures are replotted.

2. Uncertainty and effect of detection thresholds of PSMs.

(1) The neutral nucleation fractions are derived at 1.2 nm, 1.7 nm, and 2.2 nm threshold. It appears that these values are cluster mobility diameters. Please provide corresponding mass diameters and rough numbers of HOM and H2SO4 molecules in the clusters.

(2) Page 9, line 11. "... we accounted for this by increasing the detection thresholds by 0.3 nm above their calibration values". What do you mean here? So the given 1.2 nm threshold is actually 1.5 nm?

(3) Page 9, line 15. "...should be noted that the reported diameters could be systematically underestimated by up to 0.5 nm." So the given 1.2 nm threshold could actually be 1.7 nm? The actual sizes are important as charged fractions decrease quickly with cluster sizes. Please more specific so readers can better understand the results.

(4) Page 9, line 18. "... the cut-off diameter for ions can be up to 0.5 nm smaller than for neutral particles". Does this imply that the results for 1.7 nm could actually be those

for 1.2 nm?

This manuscript focuses on the results for clusters of 1.2 nm, 1.7 nm, and 2.2 nm. It appears the uncertainty in the cluster sizes detected could be up to 1 nm (see above comments), comparable to the size range of clusters analyzed here (1.2 nm - 2.2 nm). The authors need provide a more in-depth discussion on how this uncertainty might influence the results presented and their conclusions.

3. Based on results given in the manuscript (Figures 2-9), the role of ions in nucleation depends on multiple parameters. To help interested readers to better understand the results presented in various Figures which focus on the dependence of neutral fractions on certain parameter(s), I strongly suggest that the authors provide a table to list all controlling parameters measured (T, [cluster ions], [HOM], [H2SO4], [NO], [NH3], [MT], PS beam intensity) as well as derived Jn, Jiin, Jrec, and Jtot (Fig. 1) at the three thresholds for all data points (or cases) presented in Figure 2. The table can be provided as supplementary material. Such a table will also fulfill the ACP requirement with regard to the availability of underlying data (https://www.atmospheric-chemistry-and-physics.net/about/data_policy.html):

"Authors are required to provide a statement on how their underlying research data can be accessed. This must be placed as the section "Data availability" at the end of the manuscript before the acknowledgements. Please see the manuscript composition for the correct sequence. If the data are not publicly accessible, a detailed explanation of why this is the case is required.

The best way to provide access to data is by depositing them (as well as related meta-data) in reliable public data repositories, assigning digital object identifiers, and properly citing data sets as individual contributions. If different data sets are deposited in different repositories, this needs to be indicated in the data availability section."

4. Page 10, lines 8-9. How much can the lower temperature affect the HOMs production rate? Also see comment 1 above.

[Figure]

5. Page 10, line 22. What are the possible reasons that NO affects neutral fraction?

6. It appears that [NH3] in System IV ranged from 178 ppt to 1971 ppt (Table 2). Did you observe any effects of [NH3] on neutral fraction? How does the effect of [NH3] compare to that of [NO]?

7. Page 11, lines 17-19: "We compared the 1.2 nm formation rates in CLOUD with the nucleation rates of 1.5 nm particles (Kulmala et al., 2013), and the recombination rates of 1.5–1.7 nm particles (Kontkanen et al., 2013). In addition, we compared the formation rates of 1.7 nm particles in CLOUD with the formation rates at 2 nm from Manninen et al. (2009)." If you compared the 1.7 nm formation rates in CLOUD with the nucleation rates of 1.5 nm particles (Kulmala et al., 2013) and the formation rates of 2.2 nm particles in CLOUD with the formation rates at 2 nm from Manninen et al. (2009), would the conclusion change? How the possible uncertainty the PSM thresholds (see comment 2 above) may affect the comparisons and conclusions?

Minor comments

1. Page 7, lines 10-11. A factor of two error: Please provide the possible sources of errors and relevant references.

2. Page 7, line 17. NH3 detection limit: Any reference?

3. Figure 1. Which system in Table 1 does this example case correspond to? If system III or IV, what was the concentrations of NO and/or NH3?

4. Per ACP Data Policy (https://www.atmospheric-chemistry-and-physics.net/about/data_policy.html), please provide a "Statement on the availability of underlying data" (also see main comment #3).

---

## Author Comment (AC1) · 6 Nov 2017

We thank the anonymous referees for their comments, which have helped us improve the manuscript. Find below our detailed response to all of the comments.

**Anonymous Referee #1**

Ion-induced nucleation has been widely accepted as an important source of new-particle formation, as well as a famous argument for its ratio in atmosphere. This manuscript aims to reveal the ion role via a well-designed experiment at the CERN CLOUD chamber with novel characteristic instruments. This study presents very important experimental data to support the enhancement of ions in the nucleation process. The first time the ion contribution has been examined in such detail. In general, this manuscript was well-organized and the main conclusions will help improve the current understanding of new-particle formation. This manuscript should be published in ACP. I suggest a little more discussion and analysis to clarify the details behind the presented results.

Specific comments:

1. Page 9, line 30: ": : :a linear decay." If the charged fraction was a linear decay in the averaged experimental data, the author possibly could further estimate the charged fraction in a lower diameter, such as < 0.5 nm, which can reveal the role of ions in the initial stage of nucleation.

That would be certainly interesting, however, we only evaluated fractions at three diameters, therefore we cannot do an extrapolation towards smaller diameters with good confidence. It is also difficult to estimate what is the smallest diameter which is physically relevant in each chemical system and chamber conditions (so-called critical diameter).

2. Page 10, line 7-10: the authors should give more detail discussion to explain the decrease of charged fraction of nucleation rate at higher temperature (figure 3 a).

Figure 3a shows that the neutral fraction of nucleation rate is lower at higher temperatures, meaning that charged fraction actually increases towards higher temperatures. At higher temperatures, there seems to also be a stronger dependency on MT concentrations. These observations can most probably be explained by the fact that neutral clusters are more stable at low temperatures, and therefore charge is not as critical in stabilizing the clusters as at warmer temperature. The temperature also affects the concentration and composition of HOMs formed from monoterpene oxidation. To clarify these points, we added a sentence of the cluster stability, and a reference to a recent paper by Frege et al. (2017), exploring the temperature dependency of HOM formation. However, the exact temperature dependency of nucleation rates from pure biogenic nucleation is subject to another study.

The revised text now reads:

"At low temperatures, all HOM species have reduced volatility and so a larger fraction can participate in particle nucleation and growth - although this is partially compensated by the slower production rate of HOMs. Temperature also affects the composition and stability of formed HOMs clusters (Frege et al., 2017). As a result, the neutral fraction at a given MT concentration is higher at lower temperatures (Figs. 3a and 3b)."

3. Page 10, line 10-12: since the ion-ion recombination increases at a larger size (such as 2.2 nm), the authors possibly can make estimation to get the ion-induced fraction at the initial nucleation stage at a molecular level. The ion contribution of nucleation at molecular level will be toward the final answer for the role of ion-induced nucleation.

Yes, we agree, however our data doesn't allow those conclusions (see answer to comment #1). It should also be pointed out that the lowest size used in this study is already at or close to molecular level (see e.g. Ehn et al., 2011, and our response to the questions of referee 2).

4. Page 10, line 21-25: the author should give more detail description on the characteristics and roles of NOx in the nucleation process to distinguish the system II and III.

The effect of NOx on HOM formation and following new particle formation in these experiments is subject to another study currently in preparation, so we do not want to go into detail here. However, it is known that NOx reduces the formation of particles and SOA from organic precursors (e.g. Wildt et al., 2014), so it is possible that it would also influence the ion-induced fraction, which is why we wanted to study systems II and III separately.

We revised the text to read:

"After addition of NO (system III) to study the possible effect of $NO_x$ on new particle formation, the gas mixture comprised monoterpenes, sulfuric acid and nitrogen oxides (NO and $NO_2$). $NO_x$ are found to decrease the particle formation rates from monoterpene oxidation in previous studies (Wildt et al., 2014)."

5. Page 10 line 29 – page 11 line 8, last paragraph: ammonia was added in the system IV to reproduce an environmental simulation. Since the ammonia ion is easier to carry positive charges, its role in the nucleation was described as a help to stabilize the sulfuric acid. I think the authors possibly could present a more detail explanation as the ion-ion recombination.

It is known that nucleation from sulfuric acid is greatly enhanced by ammonia (Kirkby et al., 2011) or other base molecules, like amines. The charge-enhancement of nucleation gets smaller when ammonia is added, mainly due to the base molecules stabilizing the neutral sulfuric acid clusters, so that the charge is not anymore needed for stabilization. The chemical composition of the nucleating molecules is relevant here, so this cannot be described only as recombination process (any positive ion would not be enough). Although the multi-component system described in this paper is more complicated than the pure acid-base system, we see a similar effect (reduction of the importance of the ion-induced nucleation) after addition of ammonia, and we can assign it to the same mechanism (enhancement of neutral nucleation relative to ion-induced).

We modified the text in the following way:

"In this multi-component system, ammonia helps to stabilize the sulfuric acid so that the neutral fraction of particle formation at 1.2 nm and 5°C (Fig. 6a) is larger towards lower MT and $H_2SO_4$ concentrations than seen in Fig. 5a (for $H_2SO_4 > 3 \times 10^6$ cm$^{-3}$). We speculate that this is due to a similar base-stabilization mechanism, as observed in Kirkby et al. (2011) for a ternary sulfuric acid-water-ammonia system, although the multi-component system studied here is more complicated than pure acid-base systems. Ions are still important in stabilizing the particles at warmer temperatures (Fig. 6a, 25°C)."

**Anonymous Referee #2**

The authors reported results from experiments at CLOUD 5 on four systems of different chemical compositions involving monoterpenes, sulfuric acid, nitrogen oxides, and ammonia. With instrument setup consisting of two nano-particle counters, one of them equipped with an ion filter, the authors were able to investigate the effect of ions on nucleation and measure the progressive neutralization due to ion-ion recombination as clusters grow. The measurements indicate that ions enhance the nucleation process when the charge is necessary to stabilize newly formed clusters, and a large fraction of the clusters carried a charge at 1.2 nm diameter but most of these charged clusters were largely neutralized before they grew to 2.2 nm. The authors also compared CLOUD measurements to atmospheric observations at SMEAR II, Hyytiälä, Finland.

The measurements and data analysis presented in this manuscript are important to better understand the role of ions in new particle formation under different conditions. The manuscript is well within the scope of ACP. Some clarifications and additional details are needed to further improve the manuscript. I recommend the publication of this manuscript in ACP after the following comments are addressed.

Main comments

1. Figures 3-6. These figures present neutral fraction of particle formation rates versus [MT] or [MT] combined with other parameters ([H2SO4], [NO], [Cluster ions]). It is established that HOMs (from the oxidation of MT) are involved in the nucleation and/or growth of clusters. Since [HOMs] are measured (page 7) and it appears that [HOMs] are affected by other parameters such as temperature (lines 7-9, page 10), I think that the results will be more straightforward and easier to understand if [MT] in x-axis is replaced with measured [HOMs] and the figures are replotted.

> The referee is correct in saying that HOMs, or rather a subset of HOMs, are most probably the molecules finally responsible for nucleation and growth. However, there is evidence that not all HOMs are extremely-low-volatile and can participate in nucleation and initial growth (Tröstl et al., 2016). The HOM volatility distribution is further modified by temperature (Frege et al., 2017) and NOx (manuscript in preparation). Investigating exactly which subset of HOMs is nucleating in each of the studied systems is beyond the scope of this study. Therefore we prefer to continue using [MT] in our figures. We want to also point out that VOC concentration data is much more readily available from the atmosphere than [HOM] data, so this makes it easier to compare the values presented here to ambient observations.

2. Uncertainty and effect of detection thresholds of PSMs.

(1) The neutral nucleation fractions are derived at 1.2 nm, 1.7 nm, and 2.2 nm threshold. It appears that these values are cluster mobility diameters. Please provide corresponding mass diameters and rough numbers of HOM and H2SO4 molecules in the clusters.

> The mobility diameter $d_p$ = 1.5 nm corresponds to a mass diameter of $d_m$ = 1.2 nm, and approximately HOM di- or trimer, or 8 $H_2SO_4$ molecules. For $d_p$ = 2.0 nm, $d_m$ = 1.7 nm, approx. 7 HOM molecules, or 22 $H_2SO_4$ molecules; $d_p$ = 2.5 nm, $d_m$ = 2.2 nm, approx. 16 HOM molecules, or 48 $H_2SO_4$ molecules.

> We added the information for the smallest size to section 2.4 Data analysis:

"We calculated formation rates at the mobility diameters of 1.5, 2.0, and 2.5 nm, which correspond to mass diameters of about 1.2, 1.7, and 2.2 nm. The size of the smallest detected clusters is similar to HOM di- or trimers, or eight sulfuric acid molecules."

(2) Page 9, line 11. ": : : we accounted for this by increasing the detection thresholds by 0.3 nm above their calibration values". What do you mean here? So the given 1.2 nm threshold is actually 1.5 nm?

(3) Page 9, line 15. ": : :should be noted that the reported diameters could be systematically underestimated by up to 0.5 nm." So the given 1.2 nm threshold could actually be 1.7 nm? The actual sizes are important as charged fractions decrease quickly with cluster sizes. Please more specific so readers can better understand the results.

Combined answer to 2 previous comments:

The PSMs were calibrated before the campaign using size-selected charged tungsten oxide (WO$_x$) particles. Because it is known that organic particles tend to activate later in DEG-vapor than inorganic ions (cut-off is shifted to higher diameter; Kangasluoma et al., 2014), we originally shifted the cut-off by 0.3 nm based on literature values (these are the diameters given originally in the manuscript) and estimated the uncertainty to be 0.5 nm due to composition.

However, to reduce the uncertainty caused by the composition effect, we now compared the measurements in CLOUD with the PSM at different saturator flow rates (cut-off sizes) to the different size bins of the NAIS, which are not affected by the composition, as the NAIS detects the size based on electrical mobility directly. The NAIS is shown in many studies to detect peak ion mobility reliably (Wagner et al., 2016). This way we could 're-calibrate' the saturator flow - cut-off size relation of the PSM for exactly the type of aerosol and chamber conditions we used in this study. As mentioned in the paper, the comparison showed that the original estimate of 0.3 nm shift was too small. Based on the comparison, we re-selected the PSM data and our lowest diameter is now 1.5 nm. To retain the size of diameter steps, we also shifted the other cut-off diameters and re-calculated the J-values and ratios for these diameters, which are now 1.5, 2.0, and 2.5 nm. We estimate the new uncertainly to be ca. +/- 0.2 nm, based on NAIS size bin resolution and variability between different runs, but we believe there is no systematic underestimation due to composition anymore.

We modified the text in the following way:

"One source of uncertainty is the composition dependency of the detection thresholds of the PSMs. The instruments were calibrated using tungsten oxide particles before the measurement campaign. However, a higher detection threshold has been reported for organic particles (Kangasluoma et al., 2014). To account for this we compared the cut-off diameters of the PSM to the size bins of the NAIS in each chemical system used here, and chose the diameters based on this comparison. The NAIS is insensitive to composition as it detects the size based on ion mobility, and the size accuracy has been verified in laboratory calibrations (Wagner et al., 2016). The remaining uncertainty is in the order of +/- 0.2 nm based on limited size bin resolution and run-to-run variability."

(4) Page 9, line 18. ": : : the cut-off diameter for ions can be up to 0.5 nm smaller than for neutral particles". Does this imply that the results for 1.7 nm could actually be those for 1.2 nm? This manuscript focuses on the results for clusters of 1.2 nm, 1.7 nm, and 2.2 nm. It appears the uncertainty in the cluster sizes detected could be up to 1 nm (see above comments), comparable

to the size range of clusters analyzed here (1.2 nm - 2.2 nm). The authors need provide a more in-depth discussion on how this uncertainty might influence the results presented and their conclusions.

Even though we narrowed down the uncertainty due to composition, the uncertainty due to electric charge persist, as all available reference instruments for the PSM rely on detecting charged particles. Using particles neutralized after size-selection, Kangasluoma et al. (2016) managed to show that the uncertainty due to charge is generally smaller than the uncertainty due to composition. Neutral particles require higher supersaturations to be activated than charged ones, so this effect would mean that the cut-off size for neutral particles is higher than the values given by calibration. The difficulty comes when measuring the total population with variable fraction of ions, as the effective cut-off size would be something between the cut-off size for neutral and charged particles. It must be pointed out that we normally do not see a bi-modal activation curve during nucleation in CLOUD, so this points towards the difference being relatively small.

To estimate how the charge uncertainty would affect the ion fraction we present here a rough calculation (assuming 0.2 nm difference in activation of neutral and charged particles):
PSMt measures $J\pm(1.5nm) + Jn(1.7nm)$
PSMn measures $Jn(1.7nm)$
Resulting neutral fraction at 1.5nm: $Jn(1.7nm) / ( J\pm(1.5nm) + Jn(1.7nm) )$ instead of $Jn(1.5nm) / ( J\pm(1.5nm) + Jn(1.5nm) )$ as it should be, where Jn is the neutral and $J\pm$ ion formation rates. It can be seen that the charge effect results in a possible overestimation of the contribution of ions, since $Jn(1.5nm) > Jn(1.7nm)$. How big the overestimation exactly is depends on the charged fraction of the particles (how big $J\pm$ is compared to Jn) and the growth rate and losses (affecting how much larger J1.5 is compared to J1.7). All of these variables vary depending on chamber conditions, chemical system and precursor concentrations. Therefore, what we can say, is that we give an upper estimate for the charged fractions.

We added this sentence for clarification:

"Although we do not expect this charge effect to be significant in our study, we want to point out that the reported charged fractions represent upper-limit estimates."

3. Based on results given in the manuscript (Figures 2-9), the role of ions in nucleation depends on multiple parameters. To help interested readers to better understand the results presented in various Figures which focus on the dependence of neutral fractions on certain parameter(s), I strongly suggest that the authors provide a table to list all controlling parameters measured (T, [cluster ions], [HOM], [H2SO4], [NO], [NH3], [MT], PS beam intensity) as well as derived Jn, Jiin, Jrec, and Jtot (Fig. 1) at the three thresholds for all data points (or cases) presented in Figure 2. The table can be provided as supplementary material. Such a table will also fulfill the ACP requirement with regard to the availability of underlying data (https://www.atmospheric-chemistry-andphysics.net/about/data_policy.html):
"Authors are required to provide a statement on how their underlying research data can be accessed. This must be placed as the section "Data availability" at the end of the manuscript before the acknowledgements. Please see the manuscript composition for the correct sequence. If the data are not publicly accessible, a detailed explanation of why this is the case is required. The best way to provide access to data is by depositing them (as well as related metadata) in reliable public data repositories, assigning digital object identifiers, and properly citing data sets as individual contributions. If different data sets are deposited in different repositories, this needs to be indicated in the data availability section."

We added the requested statement on data availability. The data necessary to reproduce the presented graphs (J ratios and corresponding chamber conditions) will be available from Zenodo at DOI 10.5281/zenodo.1033853. However, we prefer not to disclose separate nucleation rates, in order not to jeopardize the publication of other papers that are currently in preparation, studying J vs. HOMs in different systems.

4. Page 10, lines 8-9. How much can the lower temperature affect the HOMs production rate? Also see comment 1 above.

The exact values are not known (again, this is a subject of a future study from the CLOUD group), Quantitative HOM measurements at different temperatures are not straightforward. Qualitatively we can say that the production rates are lower at low temperatures, and the paper by Frege et al. (2017) explores how the temperature affect the composition of HOMs.

5. Page 10, line 22. What are the possible reasons that NO affects neutral fraction?

See our answer to referee 1, question 4. Presence of NO affects HOM composition resulting in less stable clusters, therefore ions are more important for stability and the neutral fraction decreases. (MS in prep. Yan et al.)

6. It appears that [NH3] in System IV ranged from 178 ppt to 1971 ppt (Table 2). Did you observe any effects of [NH3] on neutral fraction? How does the effect of [NH3] compare to that of [NO]?

We didn't find a dependence on $NH_3$ other than the increased neutral fraction as soon as NH3 was present. In earlier studies it has been noticed that the effect of ammonia (or a base) on J saturates above certain concentration (Kirkby et al., 2011), which could explain the result. However, most of our experiments here were done at about constant $NH_3$ concentration, so this effect was not studied in detail here.

7. Page 11, lines 17-19: "We compared the 1.2 nm formation rates in CLOUD with the nucleation rates of 1.5 nm particles (Kulmala et al., 2013), and the recombination rates of 1.5–1.7 nm particles (Kontkanen et al., 2013). In addition, we compared the formation rates of 1.7 nm particles in CLOUD with the formation rates at 2 nm from Manninen et al. (2009)." If you compared the 1.7 nm formation rates in CLOUD with the nucleation rates of 1.5 nm particles (Kulmala et al., 2013) and the formation rates of 2.2 nm particles in CLOUD with the formation rates at 2 nm from Manninen et al. (2009), would the conclusion change? How the possible uncertainty the PSM thresholds (see comment 2 above) may affect the comparisons and conclusions?

If we shifted our comparison as suggested, the conclusions would not change, because the trends are similar for all studied diameters. Moreover, since we calibrated our cut-off sizes and our diameter scale has shifted, the comparison diameters match well. The uncertainty in the PSM threshold diameters was substantially reduced through the intercomparison of PSM and an ion mobility spectrometer, therefore, comparisons as well as conclusions are valid.

Minor comments

1. Page 7, lines 10-11. A factor of two error: Please provide the possible sources of errors and relevant references.

Error sources in absolute concentrations are a scale uncertainty, the charging efficiency in the ion source, a mass dependent transmission efficiency, and sampling line losses. The different contributions of those errors are discussed by Kirkby et al. (2016).

We modified the text as follows:

"Concentrations are subject to a systematic scale uncertainty, as well as uncertainties in charging efficiency in the ion source, a mass dependent transmission efficiency, and sampling line losses (Kirkby et al., 2016). The estimated error of absolute molecule concentrations is roughly a factor of two."

2. Page 7, line 17. $NH_3$ detection limit: Any reference?

Unfortunately, this information is so far only included a master thesis in German language that is not publicly available.

3. Figure 1. Which system in Table 1 does this example case correspond to? If system III or IV, what was the concentrations of NO and/or $NH_3$?

This case corresponds to system III, the concentration of NO was 0.084 ppb. We added this information to the caption.

4. Per ACP Data Policy (https://www.atmospheric-chemistry-andphysics.net/about/data_policy.html), please provide a "Statement on the availability of underlying data" (also see main comment #3).

We added the requested information.

**References**

[revised manuscript text omitted]